# Exploring the Potential of Sulfonamide-Dihydropyridine Hybrids as Multitargeted Ligands for Alzheimer’s Disease Treatment

**DOI:** 10.3390/ijms24119742

**Published:** 2023-06-04

**Authors:** Imen Dakhlaoui, Paul J. Bernard, Diana Pietrzak, Alexey Simakov, Maciej Maj, Bernard Refouvelet, Arnaud Béduneau, Raphaël Cornu, Krzysztof Jozwiak, Fakher Chabchoub, Isabel Iriepa, Helene Martin, José Marco-Contelles, Lhassane Ismaili

**Affiliations:** 1Laboratoire LINC UR 481, Pôle de Chimie Médicinale, Université de Franche-Comté, F-25000 Besançon, France; imen.dakhlaouii@gmail.com (I.D.);; 2Laboratory of Applied Chemistry, Heterocycles, Lipids and Polymers, Faculty of Sciences of Sfax, University of Sfax, B. P 802, Sfax 3000, Tunisia; 3Department of Biopharmacy, Medical University of Lublin, ul. W. Chodzki 4a, 20-093 Lublin, Poland; 4PEPITE EA4267, Université de Franche-Comté, F-25000 Besançon, France; 5Department of Organic Chemistry and Inorganic Chemistry, Universidad de Alcalá, Ctra. Madrid-Barcelona, Km. 33,6, 28871 Alcalá de Henares, Spain; 6Laboratory of Medicinal Chemistry (IQOG, CSIC), C/Juan de la Cierva 3, 28006 Madrid, Spain

**Keywords:** calcium channel antagonism, cholinesterase inhibition, Hantzsch reaction, multitarget directed ligands, Nrf2

## Abstract

Alzheimer’s disease (AD) is a multifactorial neurodegenerative disease that has a heavy social and economic impact on all societies and for which there is still no cure. Multitarget-directed ligands (MTDLs) seem to be a promising therapeutic strategy for finding an effective treatment for this disease. For this purpose, new MTDLs were designed and synthesized in three steps by simple and cost-efficient procedures targeting calcium channel blockade, cholinesterase inhibition, and antioxidant activity. The biological and physicochemical results collected in this study allowed us the identification two sulfonamide-dihydropyridine hybrids showing simultaneous cholinesterase inhibition, calcium channel blockade, antioxidant capacity and Nrf2-ARE activating effect, that deserve to be further investigated for AD therapy.

## 1. Introduction

Alzheimer’s disease (AD) is the most common cause of memory impairment and dementia in elderly people [1]. AD is characterized by a series of highly interconnected pathological processes whose main features are: the accumulation and aggregation of abnormal extracellular deposits of amyloid-beta peptide (Aβ), and the intracellular deposits named neurofibrillary tangles (NFTs), composed by aggregates of hyperphosphorylated tau protein [2].

According to the amyloidogenic pathway, two main isoforms of Aβ in AD brains, Aβ_1–40_ and Aβ_1–42_, are produced as a result of sequential proteolysis of amyloid precursor protein by β and γ-secretase [3,4]. An inadequate clearance of Aβ peptides induces their accumulation and, subsequently, the formation of senile plaques, associated with AD pathogenesis [5].

Hyperphosphorylation of tau protein occurs when there is an imbalance between phosphorylation and de-phosphorylation, regulated by various kinases and phosphatases, namely glycogen synthase kinase 3β (GSK-3β) and protein phosphatase 2A [6,7]. Hyperphosphorylated tau protein can alter cell viability and form aggregates in a self-replicating way, leading to NFTs accumulation, and neuronal toxicity [8].

These factors ultimately lead to a progressive loss of cholinergic neurons, memory and cognitive dysfunctions [9] connected with the decreased concentration of the neurotransmitter acetylcholine (ACh) [10] following its excessive degradation [11].

ACh is hydrolyzed by cholinesterases (ChEs) into choline and acetic acid at the neuronal level. There are two kinds of ChE in vertebrates: acetylcholinesterase (AChE), which is responsible for the hydrolysis of ACh in cholinergic brain synapses and neuromuscular junctions [12], and butyrylcholinesterase (BChE), which probably plays only a supporting role, hydrolyzing ACh and other esters [13].

ChE inhibitors are currently the only treatment option [14] to maintain an appropriate level of ACh.

The Aβ deposits, as well as NFTs and oxidative stress (OS), produce pathologically high concentrations of L-glutamate, resulting in the overactivation of voltage-dependent *N*-methyl-D-aspartate (NMDA) receptors. This phenomenon is called excitotoxicity [15,16] and it leads to an excessive influx of Ca^2+^ into the neurons, which affects neuroplasticity, and thus causes neuronal damage and cell death [17]. To prevent this, NMDA receptor antagonists can be used, e.g., memantine as one of the approved drugs for AD patients.

Another significant factor in the pathogenesis of AD is the OS [18]. The accumulation of reactive oxygen species (ROS) and reactive nitrogen species (RNS) inevitably leads to significant neuron damage [19]. These radical species are caused by different factors like mitochondrial dysfunction [20], altered homeostasis of metals such as copper, zinc and iron, linked to their implication in Aβ aggregation [21], neuroinflammation [22,23] or H_2_O_2_, formed during the catalytic oxidation of biogenic amines by monoamine oxidases (MAO) [24].

It is found that one transcription factor, called nuclear factor erythroid 2-related factor 2 (Nrf2), plays a key role in regulating cellular antioxidant response and, hence, in protecting cells from OS. When cells are exposed to OS, Nrf2 is activated and soon after, it translocates into the cell nucleus [25]. Once in the nucleus, Nrf2 binds to specific cis-acting regulatory enhancer element DNA sequences, known as antioxidant response elements (AREs), and activates the transcription of genes that encode antioxidant enzymes, such as superoxide dismutase, catalase, and glutathione peroxidase [26]. Due to these enzymes, ROS become neutralized, which prevents cells from damage. In addition, Nrf2 can also trigger the expression of genes involved in other cellular defence mechanisms, namely protein folding and clearance, autophagy, and DNA repair [27]. Therefore, Nrf2 activation is a potential therapeutic target for various diseases, associated with OS, including neurodegenerative diseases such as AD and Parkinson’s disease, as well as cancer, diabetes and cardiovascular diseases [28,29].

Calcium is an essential ion in signalling, gene expression, and metabolic pathways. In the brain, calcium is vital for neuronal function, synaptic plasticity, and neurotransmitter release. However, excessive intracellular calcium levels, resulting from calcium entry through L-type Ca^2+^ channels (VGCC), can be toxic to neurons and lead to cell death [30,31].

Indeed, the dysregulation of calcium can result in a wide range of pathological changes in the brain, such as impaired synaptic function, neuronal loss, mitochondrial disruption and increased inflammation. As a consequence, the following increase in β-secretase activity facilitates the formation of Aβ [32,33], which in turn opens calcium-permeable pores across the plasma membrane, causing an even greater rise in calcium levels.

Additionally, increased cytosolic calcium levels regulate protein kinase C and GSK-3β, promoting NFT formation [34], contributing to muscarinic cholinergic receptor activation and thus disrupting calcium signalling.

Consequently, targeting calcium dysregulation is one of the potential strategies for AD therapy.

Considering the above-mentioned elements regarding the pathophysiological complexity of AD, the multi-target strategy [35] constitutes an interesting approach in the search for new effective drugs. This strategy allows the development of ligands able to bind simultaneously to various enzymatic systems or receptors involved in the progression of this disease.

Consequently, several promising multi-target-directed ligands (MTDLs) have been developed by many research groups [36,37,38,39].

Our group has contributed to this field by using multicomponent reactions (MCR) for their facility of execution, time gain, flexibility and the variety of the obtained structures [40,41,42,43].

In the present paper, we describe the design, synthesis and biological assessment of the first novel MTDLs, which simultaneously target ChE inhibition, blockade of calcium channels, associated with antioxidant activity, and Nrf2 activation. These new MTDLs result from the association of sulfonamide moieties into a 1,4-dihydropyridine (1,4-DHP) core (Figure 1).

1,4-DHPs, such as nilvadipine, are privileged scaffolds in medicinal chemistry, well-known calcium channel blockers, which have been the subject of phase III clinical trials targeting mild to moderate AD [44]. Indeed, calcium channel blockers may have a neuroprotective effect by preventing Aβ peptide aggregation and NFT formation. It is, therefore, plausible that 1,4-DHPs may prevent or slow the progression of AD [45].

Sulfonamides have a wide range of biological applications for the treatment of diseases, including central nervous system disorders, such as schizophrenia, depression, dementia or AD [46], being particularly able to act as ChE inhibitors [11,47,48] and Nrf2 activators [49,50,51]. Thus, sulfonamide and its derivatives have been used in designing MTDLs against various diseases [14].

## 2. Results

### 2.1. Synthesis

The new MTDLs **4a**–**i** were prepared as shown in Figure 1. The synthetic scheme started from the Hantzsch MCR of commercially available 4-nitrobenzaldehyde (**1**) with alkyl acetoacetate, and ammonium acetate, in a mixture of EtOH/H_2_O, under microwave irradiation (MWI), to afford dialkyl 2,6-dimethyl-4-(4-nitrophenyl)-1,4-dihydropyridine-3,5-dicarboxylate **2a**–**c.** The catalytic hydrogenation of compounds **2a**–**c**, in the presence of Pd/C, in ethyl acetate, at room temperature (rt), for 24 h, afforded the corresponding amino compounds **3a**–**c.** Finally, compounds **4a**–**i** were prepared by reacting compounds **3a**–**c** with (benzene, *p*-toluene or methane)-sulfonyl chlorides in pyridine at rt (2–3 h). The resulting mixture was then purified by column chromatography using *n*-hexane/ethyl acetate as eluent to obtain the target compounds with yields ranging from 53% to 76%. All synthesized compounds were characterized by ^1^H and ^13^C NMR and elemental analysis. These data are reported in the experimental section and the Supporting Information.

### 2.2. Biological Assesment

To verify the efficacy of our design, the compounds were subjected to biological (ChE inhibition, calcium channel blockade, transcriptional activation of Nrf2) and physicochemical evaluation (antioxidant activity).

#### 2.2.1. Cholinesterases Inhibition

The ChE inhibition of compounds **4a**–**i** was evaluated using *Ee*AChE and *eq*BChE and donepezil and tacrine as references.

As indicated in Table 1, only compound **4f** exhibited *Ee*AChE inhibition with an IC_50_ equal to 12.6 μM compared to donepezil which showed an IC_50_ equal to 20.8 nM. All other compounds showed less than 50% inhibition at 10 μM. However, their IC_50_s could not be determined due to the solubility limit of these compounds in the buffer solution at very high concentrations. Regarding *eq*BChE inhibition, compounds **4a** (R_1_ = Me, R_2_ = Ph), **4d** (R_1_ = Et, R_2_ = *p*MePh), and **4f** (R_1_ = Et, R_2_ = Me) exhibited IC_50s_ equal to 5.0, 0.30 and 8.7 μM, respectively, compared to tacrine which showed an IC_50_ equal to 2.2 nM. The best compound, **4d,** is only 135-fold less active than tacrine, one of the known strongest BChE inhibitors.

Regarding the selectivity, most of the active compounds were preferential inhibitors of BChE. Compounds **4a** (R_1_ = Me, R_2_ = Ph) and **4d** (R_1_ = Et, R_2_ = *p*MePh) were totally selective to BChE, while compound **4f**, bearing a methylsulfonamide group and an ethyl ester, was active on both AChE and BChE enzymes with a selectivity value equal to 1.5.

Due to the small number of molecules active on the ChEs, it is difficult to discuss structure-activity relationships (SAR) in sufficient depth.

#### 2.2.2. Calcium Channel Inhibition

The calcium channel blockade of compounds **4a**–**i**, and nimodipine as a reference, was evaluated at 10 μM concentration and is reported in Table 1. Eight of the nine compounds showed calcium channel inhibition with values ranging from 22% for **4d** (R_1_ = Et, R_2_ = *p*MePh) to 51% for **4h** (R_1_ = *i*Pr, R_2_ = *p*MePh). The most potent compounds corresponded, in decreasing order, to **4h** with 51% and **4g** (R_1_ = *i*Pr, R_2_ = Ph) with 50% comparing thus very favourably with nimodipine (52%).

Interestingly, the three ChE inhibitors **4a** (R_1_ = Me, R_2_ = Ph), **4d** (R_1_ = Et, R_2_ = Ph) and **4f** (R_1_ = Et, R_2_ = Me) also showed calcium channel inhibition with values equal to 32%, 22% and 27%, respectively.

According to the SAR and for the same sulfonamide group, the best results were always obtained for compounds bearing the isopropyl ester group, followed by those bearing the methyl ester group, except for compound **4i**. For the same ester group, the nature of the sulfonamide scaffold does not play a significant role in the activity.

#### 2.2.3. Antioxidant Assay

The antioxidant capacity of compounds **4a**–**i** was evaluated by the oxygen radical absorbance capacity (ORAC) method [52], using melatonin as a reference. Radical scavenging activities are reported as Trolox Equivalents (TE). As shown in Table 1, all compounds exhibited antioxidant capacity with values ranging from 0.86 TE for **4g** (R_1_ = *i*Pr, R_2_ = Ph) to 3.01 TE for **4c** (R_1_, R_2_ = Me). The best compound, **4c,** is more active than melatonin which showed a 2.45 TE [53].

The three identified MTDLs **4a** (R_1_ = Me, R_2_ = Ph), **4d** (R_1_ = Et, R_2_ = Ph) and **4f** (R_1_ = Et, R_2_ = Me), showing ChE and calcium channel inhibitory activity, also exhibit antioxidant activities with ORAC values equal to 1.59, 1.67, and 1.30 TE, respectively, and can be compared favourably to melatonin. Compound **4a**, for example, is only 35% less active than melatonin.

Therefore, these three compounds were chosen to evaluate their Nrf2 transcriptional activation potencies.

#### 2.2.4. Nrf2 Transcriptional Activation Potencies of MTDLs **4a**, **4d** and **4f**

The Nrf2-ARE activating effect of selected **4a**, **4d** and **4f** was evaluated in vitro using a cell-based luciferase assay in the AREc32 cell line [29], *tert*-butylhydroquinone (TBHQ) being used as positive control.

Prior to this, the cytotoxicity of the compounds against AREc32 cells was assessed by measuring cell viability. All three compounds showed no toxicity up to 50 μM.

Then, AREc32 cells were treated with increasing concentrations of each compound (1, 5, 10, 25, 50 μM) for 24 h, and luciferase activity was then measured.

As shown in Figure 2, no significant activity was observed for compound **4d**. Interestingly, compounds **4a** and **4f** induced the Nrf2 transcriptional pathway significantly and successfully as early as 25 μM for **4a** and 50 μM for **4f**.

The concentrations required to double the specific activity (CD) of the luciferase reporter were then calculated to compare relative potencies. (Table 2). As expected, compounds **4a** and **4f** were the best compounds showing a CD value equal to 19.3 and 44.3 μM, respectively, compared to TBHQ, which showed 1.2 μM. Interestingly, compound **4a** is only 16-fold less active than TBHQ, this one being one of the most potent activators of Nrf2. Nevertheless, compound **4f**, with a CD value equal to 44.3 μM, i.e., 36-fold less active than TBHQ, shows, however, a 1.4-fold higher activity than melatonin (CD = 60 μM) [54], which is known for its ability to induce the transcriptional pathway [55].

### 2.3. Molecular Docking Studies of Compounds ***4a*** and ***4f***

To explore the possible binding modes and the interactions of compounds **4a** and **4f** with EeAChE and eqBChE, docking studies were carried out using AutoDock Vina [56] software v.1.2.0.

The 3D structure of EeAChE was retrieved from the Protein Data Bank (PDB ID: 1C2B), a single catalytic subunit of the enzyme was used, and the flexibility of eight side chains has been considered by allowing side chain flexibility during the docking.

As shown in Figure 3 the most energetically favoured binding mode places the ligand **4a** in the peripheral anionic site (PAS); therefore, no interactions with the catalytic triad residues can be established.

In the complex, **4a** adopts a folded conformation where the phenyl ring of the sulfonamide moiety is interacting with Tyr341 and Trp286 via π–π T-shaped interactions and with Tyr72 via π–π stacked interactions. The oxygen atoms of sulfonamide and ester groups create a hydrogen bond with the hydroxyl group of Tyr72 and the indole NH of Trp286, respectively. Additionally, π–π T-shaped interactions were observed between the phenyl substituent of the dihydropyridine moiety and Tyr341 (Figure 4a).

Molecular docking showed that compound **4f** is located in a different way than compound **4a** (Figure 3). The docking results for compound **4f** suggest that it can fit well in the active site of *Ee*AChE and interact with important amino acid residues. The less buried sulphonamide moiety permits the ligand to reach the bottom of the narrow gorge and to interact with two amino acids of the catalytic triad.

The methylsulfonamide moiety is pointed toward the catalytic triad residues, and it interacts via carbon hydrogen interactions with Ser203 and with His447 forming a hydrogen bond. It was also found that this moiety of the ligand is forming π–sulfur interactions with Trp86. The phenyl ring, in the middle of the gorge, interacts with Tyr337 via π–π stacked interactions. One of the ester and methyl groups is in the acyl binding pocket interacting with Phe338 and Phe297 via π-alkyl interactions and Phe295 via van der Waals interactions. These groups can also interact with two amino acids in the PAS (Tyr341 and Tyr124) (Figure 4b).

The other ester and methyl groups are in the PAS, with the oxygen atoms involved in two hydrogen bonds with Tyr124 and Tyr337. In addition, these moieties can form π-alkyl interactions with Tyr341, Tyr337 and Tyr124 (Figure 4b).

To elucidate the interactions between compounds **4a** and **4f** and *eq*BChE, we also performed docking studies into the active site of the homology-modelled *eq*BChE. Without the X-ray structure of horse BChE (*h*BChE), a homology model has been retrieved from the SWISS-MODEL Repository [57] to rationalize the experimental data.

The ligand **4a** is placed into the binding pocket of *eq*BChE, interacting with the residues involved in catalysis (CAS), with the residues in the oxyanion hole (OH), with the residues in the acyl-binding pocket (ABP) and with the residues in the PAS (Figure 5a). The protein complex with the best-docked pose of inhibitor showed that the phenyl ring in the dihydropyridine moiety is in the middle of the receptor cavity and with both the benzenesulfonamide and dihydropyridine groups located deep inside the gorge.

The dihydropyridine moiety binds in the CAS region of the enzyme. Specifically, the NH and methyl groups form a hydrogen bond and π-alkyl interactions with the catalytic triad residue His438, respectively. Besides, the dihydropyridine moiety is parallel to Trp82, establishing π–sigma and π–alkyl interactions with three methyl groups. One of the methyl groups is engaged in a network of π–alkyl and alkyl interactions with Trp430, Tyr440, Ala328, Met437 and Tyr440. The benzenesulfonamide ring is involved in π–π T–shaped interactions with Trp231 and Phe329 (ABP) and in π–alkyl interactions with Leu286 (ABP). Gly116 (OH) allowed an amide-π stacked interaction with this benzene ring. The ligand displays an additional hydrogen bond between the NH of the sulfonamide moiety and Leu285. An ester group interacts weakly with Asp70 and Tyr332, in the PAS (Figure 6a).

The docking results revealed that BChE could effectively accommodate compound **4f** inside the active site gorge (Figure 5b). The dihydropyridine moiety is oriented toward the bottom of the active site and it binds in the CAS region of the enzyme, establishing a key π-alkyl interaction with His438 and alkyl, π-alkyl and carbon–hydrogen interactions with two key amino acids of the ABP (Trp82 and Ala328). Leu285 residue and Gly116 (OH) further stabilized the position of the methylsulfonamide moiety in the active-site gorge via hydrogen bond and carbon–hydrogen interactions (Figure 6b). Besides, the esters groups display additional interactions with Tyr332 and Asp70, in the PAS (Figure 6b).

Based on docking results, compound **4a** interacted with the mouth of the active gorge of *Ee*AChE. In contrast, the most potent compound, **4f,** can fit well in the active site interacting with important amino acid residues on both pockets, the CAS and PAS. Docking studies also revealed the capability of compounds **4a** and **4f** to bind to CAS and PAS of *eq*BChE and induce its inhibitory effect.

## 3. Discussion 

In this study, our focus was on the design, synthesis, and biological evaluation of a novel class of compounds obtained by combining sulfonamide moieties with a 1,4-dihydropyridine scaffold. These compounds exhibit ChE inhibition, calcium channel blockade, Nrf2 pathway activation, and antioxidant activity, well-established therapeutic targets for AD.

We found that, without exception, all compounds exhibited antioxidant activity as measured by the ORAC test. This fluorescence-based test measures the ability of molecules to trap the radical derived from AAPH, suggesting that these molecules can donate a hydrogen atom, specifically from the DHP, as well as the ability to stabilize a free radical.

At the same time, almost all the prepared compounds showed calcium channel blockade, comparable to nimodipine used as a reference. Further investigation is ongoing in our laboratory to establish the mode of action of such compounds on calcium channel activity, as well as to develop a complete SAR profile with a larger number of analogues.

Conversely, a few compounds are active on ChE. Two compounds, 4a and 4d, showed fully selective inhibition of BChE with IC_50_ values in the micromolar range, whereas compound 4f showed dual inhibition of both ChEs.

These three selected compounds can induce the Nrf2 transcriptional pathway significantly and successfully, with compound 4a exhibiting an effect as early as 25 μM and compound 4f at 50 μM. They can therefore activate endogenous antioxidant enzymes such as superoxide dismutase, catalase, and glutathione peroxidase. This is likely due to the presence of the sulfonamide moiety and its antioxidant activities.

From this work, compounds **4a** and **4f** were identified as the first generation of MTDLs combining simultaneous ChE inhibition, calcium channel blockade, antioxidant capacity, and an activation effect of Nrf2-ARE. These compounds could serve as templates for developing more potent multitarget compounds for targets related to AD.

In addition, these molecules show great promise compared to the sulfonamides recently described in the literature [12,49]. Regarding AChE inhibition, these molecules are less active than the sulfonamides described by Yamali et al. [48], which exhibit nanomolar activities between 8 and 15 nM, but they are at least 10-fold more active than those described by the group of Enriz and Imramovsky [11], which show activities between 55 and 150 μM. For BChE, it is worth noting that the sulfonamides prepared in this study are significantly more active than those described in the literature [12]. As an example, compound **4d**, the most potent BChE inhibitor, is 173 to 1700 times more active than the sulfonamides described by Enriz and Imramovsky [11], which show IC_50_ values between 52 and 112 μM, and 23 to 70 times more active than the sulfonamides reported by Singh’s group [47], where the IC_50_ values range from 7 to 21 μM.

The binding modes of these compounds towards both ChEs were determined through molecular docking. The results showed that compound 4a interacts with the mouth of the active gorge of AChE, while the most potent compound, **4f**, fits well in the active site and interacts with important amino acid residues in both CAS and PAS. Docking studies also revealed the capability of compounds **4a** and **4f** to bind to both CAS and PAS of BChE, resulting in their inhibitory effect.

## 4. Materials and Methods

Monitoring of reaction progress was performed by analytical thin-layer chromatography (TLC) on aluminium sheets precoated with silica gel (Type 60 F254, 0.25 mm; from Merck, Darmstadt, Germany). NMR spectra were measured on a BRUCKER DRX-400 AVANCE spectrometer using dimethylsulfoxide (DMSO-d_6_) or chloroform (CDCl_3_) as solvents. Chemical shifts are expressed in parts per million (ppm) and multiplicities of ^1^H NMR signals were labelled as follows: s: singlet; d: doublet; t: triplet; q: quartet; and m: multiplet and coupling constants were expressed in hertz (Hz). Elemental analyses were performed on a Carlo–Erba CHNS apparatus. The purity of the compounds **4a**–**i** was checked and confirmed to be >95% by elemental analyses, conducted on a Carlo–Erba EA 1108. These compounds were also found to be ≥95% pure by HPLC analysis using a Hitachi Chromaster instrument equipped with a BDS Hypersil C18 column (4.6 mm × 250 mm, Ø = 5 μm). The mobile phase is a mixture of methanol—and aqueous potassium phosphate 0.1 g% and phosphoric acid at 85% at 0.05 g% (60:40, *v*/*v*), and peaks were detected at 230 nm (see chromatograms of **4a** and **4f** in the Supporting Information)

### 4.1. General Synthesis of Compounds ***2a***–***c***

A solution of 1 mmol of 4-nitro-benzaldehyde and 2.5 mmol of the appropriate alkyl acetoacetate in the presence of 1.2 mmol of ammonium acetate was heated under MWI at 150 °C for 5 min. After cooling, an ethanol/water mixture (1:1) was added and the precipitate formed was collected in solid form by filtration to give compounds **2a**–**c** with yields ranging from 75% to 96%.

#### 4.1.1. Dimethyl 2,6-Dimethyl-4-(4-nitrophenyl)-1,4-dihydropyridine-3,5-dicarboxylate (**2a**)

Yield: 96%. ^1^H NMR (400 MHz, CDCl_3_/TMS): δ 8.1 (d, *J* = 8.7 Hz, 2H), δ 7.45 (d, *J* = 8.7 Hz, 2H), 5.8 (s, 1H, NH), 5.12 (s, 1H), 3.66 (s, 6H), 2.38 (s, 6H).

#### 4.1.2. Diethyl 2,6-Dimethyl-4-(4-nitrophenyl)-1,4-dihydropyridine-3,5-dicarboxylate (**2b**)

Yield: 86%. ^1^H NMR (400 MHz, CDCl_3_/TMS): δ 8.1 (d, *J* = 8.7 Hz, 2H), δ 7.38 (d, *J* = 8.7 Hz, 2H), 5.7 (s, 1H, NH), 5.02 (s, 1H), 2.28 (s, 6H), 1.14 (t, *J* = 7.1 Hz, 6H).

#### 4.1.3. Diisopropyl 2,6-Dimethyl-4-(4-nitrophenyl)-1,4-dihydropyridine-3,5-dicarboxylate (**2c**)

Yield: 75%. ^1^H NMR (400 MHz, CDCl_3_/TMS): δ 8.12–8.07 (m, 2H), 7.49–7.45 (m, 2H), 5.75 (s, 1H, NH), 5.08 (s, 1H), 4.97 (m, 2H), 2.36 (s, 6H), 1.26 (d, *J* = 6.2 Hz, 6H), 1.13 (d, *J* = 6.2 Hz, 6H). ^13^C NMR (100 MHz, CDCl_3_): δ 166.60, 155.28, 146.30, 144.34, 129.08, 123.17, 103.50, 67.39, 40.30, 22.1, 21.85, 19.67.

### 4.2. General Synthesis of Compounds ***3a***–***c***

To a solution of compounds **2a**–**c** (1 equiv., 1 mmol) in 75 mL of ethyl acetate, 150 mg of palladium on carbon and sodium sulfate (6 equiv., 6 mmol, 2 g) were added. The reaction mixture was subjected to hydrogenation at 40 psi hydrogen pressure for 24 h. The solution was then filtered on Celite. The filtrate was concentrated and purified using flash column chromatography (EtOAc, *n*-hexane) to obtain the desired compounds **3a**–**c** in solid form.

#### 4.2.1. Dimethyl 4-(4-Aminophenyl)-2,6-dimethyl-1,4-dihydropyridine-3,5-dicarboxylate (**3a**)

Yield: 60%. ^1^H NMR (400 MHz, CDCl_3_/TMS): δ 6.97 (d, *J* = 8.7 Hz, 2H), 6.47 (d, *J* = 8.3 Hz, 2H), 5.64 (s, 1H, NH), 4.81 (s, 1H), 3.57 (s, 6H), 2.24 (s, 6H). ^13^C NMR (100 MHz, CDCl_3_): δ 168.24, 144.56, 143.83, 138.1, 128.52, 114.93, 104.2, 50.96, 38.32, 19.56.

#### 4.2.2. Diethyl 4-(4-Aminophenyl)-2,6-dimethyl-1,4-dihydropyridine-3,5-dicarboxylate (**3b**)

Yield: 62%. ^1^H NMR (400 MHz, CDCl_3_/TMS): δ 7.07–6.89 (m, 2H), 6.54–6.36 (m, 2H), 5.54 (s, 1H, NH), 4.79 (s, 1H), 4.06–3.97 (m, 4H), 2.24 (s, 6H), 1.15 (t, *J* = 7.1 Hz, 6H). ^13^C NMR (100 MHz, CDCl_3_): δ 167.8, 144.54, 143.38, 138.52, 128.88, 114.77, 104.52, 59.65, 38.66, 19.59, 14.29.

#### 4.2.3. Diisopropyl 4-(4-Aminophenyl)-2,6-dimethyl-1,4-dihydropyridine-3,5-dicarboxylate (**3c**)

Yield: 64%. ^1^H NMR (400 MHz, CDCl_3_/TMS): δ 6.98 (d, *J* = 8.4 Hz, 2H), 6.45 (d, *J* = 8.5 Hz, 2H), 5.53 (s, 1H, NH), 4.92–4.82 (m, 2H), 4.77 (s, 1H), 2.22 (s, 6H), 1.16 (d, *J* = 6.2 Hz, 6H), 1.06 (d, *J* = 6.2 Hz, 6H). ^13^C NMR (100 MHz, CDCl_3_): δ 167.35, 144.33, 143.03, 138.7, 129.06, 114.67, 104.82, 66.85, 38.32, 22.14, 21.88, 19.57.

### 4.3. General Synthesis of Compounds ***4a***–***i***

To a solution of compounds **3a**–**c** (1 mmol) in 4 mL pyridine, cooled at 0 °C, 1 mmol of sulfonylchloride derivatives was added at 0 °C and the mixture was stirred at rt for 2~3 h. Then, 5 mL of 3N HCl ethyl acetate was added at 0 °C followed by filtration to remove the pyridine hydrochloride formed. 8 mL of a 1N hydrochloric acid solution was added to the filtrate, followed by three extractions with 3 × 15 mL ethyl acetate. The organic layers were combined, dried over Na_2_SO_4_ and evaporated. The residue obtained was purified by flash column chromatography with an *n*-hexane ethyl acetate mixture to give the expected product in solid form.

#### 4.3.1. Dimethyl 2,6-Dimethyl-4-(4-(phenylsulfonamido)phenyl)-1,4-dihydropyridine-3,5-dicarboxylate (**4a**)

Yield: 76%. ^1^H NMR (400 MHz, DMSOd_6_): δ 10.17 (s, 1H, NH), 8.83 (s, 1H, NH), 7.75 (d, *J* = 7.2 Hz, 2H), 7.61 (t, *J* = 7.3 Hz, 1H), 7.54 (t, *J* = 7.4 Hz, 2H), 6.97 (d, *J* = 8.6 Hz, 2H), 6.9 (d, *J* = 8.6 Hz, 2H), 4.77 (s, 1H), 3.51 (s, 6H), 2.23 (s, 6H). ^13^C NMR (100 MHz, DMSOd_6_): δ 167.78, 146.13, 144.10, 140.44, 135.90, 133.27, 129.7, 120.33, 101.80, 51.09, 39.36, 18.63. Anal. Calcd. for C_23_H_24_N_2_O_6_S: C, 60.51; H, 5.30; N, 6.14. Found: 60.42; H, 5.33; N, 6.17.

#### 4.3.2. Dimethyl 2,6-Dimethyl-4-(4-((4-methylphenyl)sulfonamido)phenyl)-1,4-dihydropyridine-3,5-dicarboxylate (**4b**)

Yield: 70%. ^1^H NMR (400 MHz, DMSOd_6_): δ 10.1 (s, 1H, NH), 8.83 (s, 1H, NH),7.64 (d, *J* = 8.3 Hz, 2H), 7.33 (d, *J* = 8Hz, 2H), 6.97 (d, *J* = 8.6 Hz, 2H), 6.93–6.87 (m, 2H), 4.77 (s, 1H), 3.51 (s, 6H), 2.34 (s, 3H), 2.23 (s, 6H). ^13^C NMR (100 MHz, DMSOd_6_): δ 167.79, 146.13, 143.91, 143.60, 137.61, 136.05, 130.15, 128.12, 127.08, 120.09, 101.81, 51.09, 38.28, 21.43, 18.63. Anal. Calcd. for C_24_H_26_N_2_O_6_S: C, 61.26; H, 5.57; N, 5.95. Found: C, 61.21; H, 5.59; N, 5.99.

#### 4.3.3. Dimethyl 2,6-Dimethyl-4-(4-(methylsulfonamido)phenyl)-1,4-dihydropyridine-3,5-dicarboxylate (**4c**)

Yield: 67%. ^1^H NMR (400 MHz, DMSOd_6_): δ 9.56 (s, 1H, NH), 8.88 (s, 1H, NH), 7.12–7.02 (m, 4H), 4.84 (s, 1H), 3.55 (s, 6H), 2.95 (s, 3H), 2.26 (s, 6H). ^13^C NMR (100 MHz, DMSOd_6_): δ 167.83, 146.18, 144.05, 136.6, 128.24, 128.24, 120.51, 101.91, 51.14, 38.32, 31.15, 18.66. Anal. Calcd. for C_18_H_22_N_2_O_6_S: C, 54.81; H, 5.62; N, 7.10. Found: C, 54.77; H, 5.64; N, 7.14.

#### 4.3.4. Diethyl 2,6-Dimethyl-4-(4-(phenylsulfonamido)phenyl)-1,4-dihydropyridine-3,5-dicarboxylate (**4d**)

Yield: 63%. ^1^H NMR (400 MHz, DMSOd_6_): δ 10.13 (s, 1H, NH), 8.74 (s, 1H, NH), 7.77–7.72 (m, 2H), 7.58 (t, *J* = 7.3 Hz, 1H), 7.51 (t, *J* = 7.5 Hz, 2H), 7.00 (d, *J* = 8.5 Hz, 2H), 6.92 (d, *J* = 8.5 Hz, 1H), 4.75 (s, 1H), 4.15–3.79 (m, 4H), 2.23 (s, 6H), 1.05 (s, 6H). ^13^C NMR (100 MHz, DMSOd_6_): δ 167.32, 145.72, 144.7, 140.27, 135.77, 133.19, 129.58, 128.57, 127.05, 120.37, 102.16, 59.36, 38.82, 21.2, 18.63, 14.57. Anal. Calcd. for C_25_H_28_N_2_O_6_S: C, 61.97; H, 5.82; N, 5.78. Found: C, 61.92; H, 5.83; N, 5.81.

#### 4.3.5. Diethyl 2,6-Dimethyl-4-(4-((4-methylphenyl)sulfonamido)phenyl)-1,4-dihydropyridine-3,5-dicarboxylate (**4e**)

Yield: 66%. ^1^H NMR (400 MHz, DMSOd_6_): δ 10.05 (s, 1H, NH), 8.75 (s, 1H, NH), 7.62 (d, *J*= 8.2 Hz, 2H), 7.29 (t, *J* = 8.1 Hz, 2H), 6.99 (d, *J* = 8.5 Hz, 2H), 6.91 (d, *J* = 8.5 Hz, 2H), 4.71 (s, 1H), 3.96 (m, 4H), 2.31 (s, 3H), 2.23 (s, 6H), 1.05 (t, *J* = 7.1 Hz, 6H). ^13^C NMR (100 MHz, DMSOd_6_): δ 167.33, 145.72, 144.53, 143.51, 137.43, 135.92, 130.03, 128.54, 127.11, 120.16, 102.18, 59.36, 38.79, 21.37, 21.37, 18.63, 14.55. Anal. Calcd. for C_26_H_30_N_2_O_6_S: C, 62.63; H, 6.07; N, 5.62. Found: C, 62.69; H, 6.02; N, 5.60.

#### 4.3.6. Diethyl 2,6-Dimethyl-4-(4-(methylsulfonamido)phenyl)-1,4-dihydropyridine-3,5-dicarboxylate (**4f**)

Yield: 60%. ^1^H NMR (400 MHz, DMSOd_6_): δ 9.54 (s, 1H, NH), 8.8 (s, 1H, NH), 7.1 (d, *J* = 8.6 Hz, 2H), 7.04 (t, *J* = 8.6 Hz, 2H), 4.82 (s, 1H), 3.99 (m, 4H), 2.94 (s, 3H), 2.26 (s, 6H), 1.13 (t, *J* = 7.1 Hz, 6H). ^13^C NMR (100 MHz, DMSOd_6_): δ 167.33, 145.81, 144.49, 136.51, 128.58, 127.11, 120.39, 102.21, 59.46, 38.69, 18.69, 14.66. Anal. Calcd. for C_20_H_26_N_2_O_6_S: C, 56.86; H, 6.20; N, 6.63. Found: C, 56.92; H, 6.18; N, 6.57.

#### 4.3.7. Diisopropyl 2,6-Dimethyl-4-(4-(phenylsulfonamido)phenyl)-1,4-dihydropyridine-3,5-dicarboxylate (**4g**)

Yield: 58%: ^1^H NMR (400 MHz, DMSOd_6_): δ 10.08 (s, 1H, NH), 8.65 (s, 1H, NH), 7.73–7.68 (m, 2H), 7.61–7.55 (m, 1H), 7.52–7.45 (m, 2H), 7.00–6.98 (m, 2H), 6.91–6.89 (m, 2H), 4.79–4.74 (m, 2H), 4.68 (s, 1H), 2.2 (s, 6H), 1.13 (d, *J* = 6.2 Hz, 6H), 0.94 (d, *J* = 6.2 Hz, 6H). ^13^C NMR (100 MHz, DMSOd_6_): δ 166.86, 145.35, 145.01, 140.12, 135.66, 133.17, 129.54, 128.84, 127.06, 120.38, 102.49, 66.39, 60.22, 39.12, 22.30, 21.97, 18.62. Anal. Calcd. for C_27_H_32_N_2_O_6_S: C, 63.26; H, 6.29; N, 5.46. Found: C, 63.34; H, 6.25; N, 5.42.

#### 4.3.8. Diisopropyl 2,6-Dimethyl-4-(4-((4-methylphenyl)sulfonamido)phenyl)-1,4-dihydropyridine-3,5-dicarboxylate (**4h**)

Yield: 55%. ^1^H NMR (400 MHz, DMSOd_6_): δ 10.00 (s, 1H, NH), 8.65 (s, 1H, NH), 7.58 (d, *J* = 8.3 Hz, 2H), 7.28 (d, *J* = 8Hz, 2H), 6.98 (d, *J* = 8.6 Hz, 2H), 4.79–4.74 (m, 2H), 4.68 (s, 1H), 2.31 (s, 3H), 2.2 (s, 6H), 1.13 (d, *J* = 6.2 Hz, 6H), 0.94 (d, *J* = 6.2 Hz, 6H). ^13^C NMR (100 MHz, DMSOd_6_): δ 166.87, 145.34, 144.82, 143.45, 137.29, 135.83, 129.98, 128.8, 127.12, 120.19, 102.5, 66.4, 39.09, 22.29, 21.93, 21.39, 18.62. Anal. Calcd. for C_28_H_34_N_2_O_6_S: C, 63.86; H, 6.51; N, 5.32. Found: C, 63.91; H, 6.49; N, 5.30.

#### 4.3.9. Diisopropyl 2,6-Dimethyl-4-(4-(methylsulfonamido)phenyl)-1,4-dihydropyridine-3,5-dicarboxylate (**4i**)

Yield: 60%. ^1^H NMR (400 MHz, DMSOd_6_): δ 9.52 (s, 1H, NH), 8.72 (s, 1H, NH), 7.1 (d, *J* = 8.6 Hz, 2H), 7.04 (d, *J* = 8.6Hz, 2H), 4.82 (m,2H), 4.78 (s,1H), 2.91 (s,3H), 2.24 (s,6H), 1.21–1.15 (d, *J* = 6.2 Hz, 6H), 1.05 (d, *J* = 6.2 Hz, 6H). ^13^C NMR (100 MHz, DMSOd_6_): δ 166.94, 145.49, 144.76, 143.76, 136.44, 128.83, 120.31, 102.53, 66.51, 38.93, 22.34, 22.08, 18.69. Anal. Calcd. for C_22_H_30_N_2_O_6_S: C, 58.65; H, 6.71; N, 6.22. Found: C, 58.62; H, 6.70; N, 6.24.

### 4.4. Biological Evaluation

#### 4.4.1. *Ee*AChE and *eq*BChE

ChE inhibition of compounds **4a**–**i** was performed according to Ellman’s method [58] using purified *Ee*AChE or *eq*BChE. The reaction was performed in a final volume of 3 mL of phosphate-buffered solution (0.1 M) at pH = 8.0, containing 2625 μL of 5,5′-dithiobis-2-nitrobenzoic acid (0.35 mM), 29 μL of *Ee*AChE (0.035 U/mL) or 60 μL (0.05 U/mL) of *eq*BChE, and 3 μL (0.1–10 μM, final concentrations) of the test compounds. After a pre-incubation period of 10 min, 105 μL of acetylthiocholine iodide (0.35 mM) or 150 μL of butyrylthiocholine (0.5 mM) was added and left under stirring for an additional 15 min of incubation. The absorbances were then measured at 412 nm in a spectrophotometric plate reader (iEMS Reader MF, Labsystems, Vantaa, Finland). IC_50_ values were determined graphically from log concentration–% inhibition curves, using GraphPad Prism 5.0 software, Dotmatics Boston, MA, USA.

#### 4.4.2. Calcium Channel Inhibition

Assessment of calcium channel blockade of compounds **4a**–**i** was performed according to the previously described protocol using the FLIPR calcium indicator 6 [59]. Briefly, FLIPR-loaded SH-SY5Y cells were exposed for 10 min to a 10 μm concentration of the tested compounds and to nimodipine used as a reference. 0.1% DMSO was used as a vehicle. The fluorescence change from calcium flux induced with KCl and CaCl2 (90 and 5 mM, respectively) was then recorded (λEx = 485 nm; λEm = 525 nm). Data were collected in three independent experiments with eight technical replicates per experiment. Outliers detected by the Grubbs test were excluded from further analysis.

#### 4.4.3. Oxygen Radical Absorbance Capacity Assay

Antioxidant activity of **4a**–**i** was performed by ORAC according to the protocol previously described [54]. Briefly, in a black 96-well microplate (Nunc), fluorescein and the tested compound were incubated for 15 min at 37 °C. 2,2′-Azobis(amidinopropane) dihydrochloride was then added and fluorescence was measured every minute for 1 h (λEx = 485 nm; λEm = 535 nm). All reactions were performed in triplicate, and at least three different assays were performed for each sample.

#### 4.4.4. Nrf2 Transcriptional Activation Potencies of MTDLs **4a**, **4d** and **4f**

The evaluation of Nrf2 transcriptional activation potencies of the tested compounds was performed using an Nrf2/ARE-luciferase reporter HEK293 stable cell line (Signosis, Santa Clara, CA, USA). First, we determined the cytotoxicity of the tested compounds using the MTT assay. For this purpose, the cells were seeded at a density of 2 × 10^4^ cells per well in DMEM high glucose medium supplemented with 10% FBS using transparent 96-well culture plates. After 48 h at 37 °C, in 95% air/5% CO_2_, the culture medium was replaced with fresh DMEM containing only 0.1% FBS. Different concentrations of the tested compounds or DMSO (0.1%) were added to this culture medium. After 24 h of incubation with the tested compounds, the per cent of cell viability was measured. All reactions were performed in duplicate and repeated in at least four cell cultures.

The evaluation of Nrf2 transcriptional activation was then performed using non-cytotoxic concentrations of the tested compounds according to the protocol previously described [60]. Briefly, the cells were seeded as described for the MTT assay, except that white 96-well culture plates were used. After 48 h of incubation, the cells were treated with the tested compounds or DMSO (0.1%). After 24 h of treatment, luciferase activity was measured using the Bright–Glo Luciferase Assay System (Promega, Madison, WI, USA) according to the manufacturer’s instructions. All reactions were performed in duplicate and repeated in at least four different cultures.

#### 4.4.5. Molecular Docking of Compounds **4a** and **4f** into *Ee*AChE and *eq*BChE

Compounds **4a** and **4f** were assembled within Discovery Studio (DS, version 2022), software package, using standard bond lengths and bond angles. The molecular geometries of **4a** and **4f** were energy-minimized using the adopted-based Newton-Raphson algorithm with the CHARMm force field [61]. Structures were considered fully optimized when the energy changes between iterations were less than 0.01 kcal/mol [62]. The ligand was set up for docking with the help of AutoDockTools (ADT; version 1.5.7) to define the torsional degrees of freedom to be considered during the docking process. All the acyclic dihedral angles in the ligand were allowed to rotate freely.

The 3D coordinates of *Ee*AChE (PDB ID: 1C2B), were obtained from the PDB. Then, the water molecules, heteroatoms, co-crystallized solvent and ligand were removed. Proper bonds, bond orders, hybridization and charges were assigned using the protein model tool in the DS software package Version 2022. AutoDockTools (ADT; version 1.5.7) was used to add hydrogens and partial charges for proteins and ligands using Gasteiger charges. To give flexibility to the binding site, eight residues (Trp286, Tyr124, Tyr337, Tyr72, Asp74, Thr75, Trp86 and Tyr341) lining the AChE binding site were allowed to move using the AutoTors module.

The search space was defined as a box with the following parameters: size x = 60, size y = 60, size z = 72 with grid points separated 1 Ǻ and centred at the middle of the protein (x = 21.5911; y = 87.752; z = 23.591).

The *h*BChE model has been retrieved from the SWISS-MODEL Repository [57]. A putative three-dimensional structure of eqBChE has been created based on the crystal structure of *h*BChE (PDB ID: 2PM8), these two enzymes exhibited 89% sequence identity. Initial protein preparation and docking calculations were performed following the same protocol described for *Ee*AChE. A cube of 75Ǻ with grid points separated by 1 Ǻ, was positioned in the middle of the protein (x = 29.885; y = −54.992; z = 58.141).

Docking calculations were performed with the program Autodock Vina v.1.2.0. [56] as *blinds* dockings where the num_modes was set to 40 and the other parameters were left as default parameters Finally, the most favourable conformations based on the binding energy were selected for analyzing the interactions between the enzymes and inhibitors, using DS.

## Data Availability

Samples of the compounds are available from the authors.

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
