# Peer review of "Exploring the Potential of Sulfonamide-Dihydropyridine Hybrids as Multitargeted Ligands for Alzheimer’s Disease Treatment"

_ijms, 2023, doi:10.3390/ijms24119742_

Round 1

Reviewer 1 Report

Review report

Journal: IJMS (ISSN 1422-0067)

Manuscript ID: ijms-2353690

Type: Article

Title:Exploring the Potential of Sulfonamide-Dihydropyridine Hybrids as

Multitargeted Ligands for Alzheimer's Disease Treatment”.

Authors: Imen Dakhlaoui, Paul J Bernard, Diana Pietrzak, Alexey Simakov,

Maciej Maj, Bernard Refouvelet, Arnaud Béduneau, Raphaël Cornu, Krzysztof

Jozwiak, Fakher Chabchoub, Helene Martin, José Marco-Contelles, Lhassane

Ismaili *

Section: Molecular Pharmacology

General

In this manuscript the authors explore multi-target directed ligands (MTDLs) as a promising therapeutic strategy for Alzheimer's disease. To this end, the authors designed and synthesised novel MTDLs using cost-effective procedures. The new MTDLs targeted calcium channel blockade, cholinesterase inhibition and antioxidant activity. The study identified two promising MTDLs, which they named 4a and 4f, that showed cholinesterase inhibition, calcium channel blockade, antioxidant capacity and Nrf2-ARE activating effect. According to the authors, these two MTDLs are considered potential candidates for further research in AD therapy.

Major points

The manuscript details well-executed and high-quality experiments. The results are clear and convincing. However, I consider that the primary weakness of this article lies in its writing, composition, and interpretation of the results. The observations are detailed below:

1.    The conclusion of the manuscript is remarkably poorly elaborated. There is no discussion of possible mechanisms of action of the chosen compounds, no discussion of the disadvantages of the methodology and no comparison of the results with other published work. The presented conclusion represents only a summary of the detailed data in the results section, which does not represent an in-depth analysis of the data found. I strongly suggest revising this section, taking into account the aforementioned aspects.

Minor points

2.    The authors mention in line 17 that "there is still no effective therapy". In fact, what exists today are effective therapies, however, these therapies do not solve the basic problem (the cause) of Alzheimer's disease, so a correct expression would be "there is still no cure for the disease". I suggest rewriting that line.

3.    Mentioning abbreviations in the abstract (such as 4a and 4f) in line 22 is not appropriate, as they are not yet detailed. I suggest using a generalised name that gives an indication of their chemical nature.

4.    In line 98 the authors state: ".... the multi-target strategy seems to be the most appropriate for finding new drugs...". This seems a bit contradictory, since despite the extensive literature on the development and testing of multi-target drugs; and the extensive coverage the authors gave to the multiple causes of Alzheimer's disease, it is not entirely appropriate to mention that this (multi-target) strategy is the "most appropriate". That is contrary to the evidence. There are many multi-target strategies, which have beautiful in vitro behaviour, but no in vivo perspective (because of high toxicity, solubility problems etc). Anyway, I suggest modifying that sentence and mentioning that this strategy is interesting to be explored, at the same level as other strategies. It is definitely not the "most appropriate".

5.    In line 149 the authors use the expression: "... shown GOOD EeAChE inhibition...". Question: How can the authors consider as "good" a behaviour (measured by IC50) where there is a 1000-fold difference? IC50 donepezil = 20.8 nM vs IC50 4f = 12.6 µM? Firstly, there is a lack of statistical evidence to be sure of that and secondly, the word "good" is too subjective to interpret scientific results (it seems to be more an opinion of the authors on their own results). I strongly recommend rewriting those lines. The same goes for lines 136, 154 and 195, where that word is used.

6.    In line 164 the authors mention "these molecules seem very promising, compared to the sulfonamides recently described in the literature...". This type of expression (which is subjective) should go at the end of the article and not in the first results section, as not all the findings are mentioned and their importance is not yet discussed. I recommend removing it or moving it to the "Discussion" section.

7.    At the end of section 2.2.3 (antioxidant capacity assay, line 203), the authors arbitrarily choose to evaluate the Nrf2 transcriptional activation potencies of only 3 molecules (out of the total of 8 with which the study started). Why were the others not tested? Please give an appropriate argument for your exclusion decision.

8.    In line 227 the word "powers" should be replaced by "potencies" or "capacities".

Quality of the English language is good. The text is well-written and easy to understand, with clear and concise sentences that effectively communicate the intended message. However, the wording and syntax (scientifically speaking) should be improved.

Author Response

Reviewer 1:

General

In this manuscript the authors explore multi-target directed ligands (MTDLs) as a promising therapeutic strategy for Alzheimer's disease. To this end, the authors designed and synthesised novel MTDLs using cost-effective procedures. The new MTDLs targeted calcium channel blockade, cholinesterase inhibition and antioxidant activity. The study identified two promising MTDLs, which they named 4a and 4f, that showed cholinesterase inhibition, calcium channel blockade, antioxidant capacity and Nrf2-ARE activating effect. According to the authors, these two MTDLs are considered potential candidates for further research in AD therapy.

Major points

The manuscript details well-executed and high-quality experiments. The results are clear and convincing. However, I consider that the primary weakness of this article lies in its writing, composition, and interpretation of the results. The observations are detailed below:

  1. The conclusion of the manuscript is remarkably poorly elaborated. There is no discussion of possible mechanisms of action of the chosen compounds, no discussion of the disadvantages of the methodology and no comparison of the results with other published work. The presented conclusion represents only a summary of the detailed data in the results section, which does not represent an in-depth analysis of the data found. I strongly suggest revising this section, taking into account the aforementioned aspects.

Answer: According to the reviewer’s remark the conclusion has been rewritten

Minor points

  1. The authors mention in line 17 that "there is still no effective therapy". In fact, what exists today are effective therapies, however, these therapies do not solve the basic problem (the cause) of Alzheimer's disease, so a correct expression would be "there is still no cure for the disease". I suggest rewriting that line.

Answer: According to the reviewer’s remark this sentence was rewritten

  1. Mentioning abbreviations in the abstract (such as 4a and 4f) in line 22 is not appropriate, as they are not yet detailed. I suggest using a generalised name that gives an indication of their chemical nature.

Answer: Good suggestion, accordingly, the “MTDL 4a and 4f” were replaced by “two sulfonamide-dihydropyridine hybrids”

  1. In line 98 the authors state: ".... the multi-target strategy seems to be the most appropriate for finding new drugs...". This seems a bit contradictory, since despite the extensive literature on the development and testing of multi-target drugs; and the extensive coverage the authors gave to the multiple causes of Alzheimer's disease, it is not entirely appropriate to mention that this (multi-target) strategy is the "most appropriate". That is contrary to the evidence. There are many multi-target strategies, which have beautiful in vitro behaviour, but no in vivo perspective (because of high toxicity, solubility problems etc). Anyway, I suggest modifying that sentence and mentioning that this strategy is interesting to be explored, at the same level as other strategies. It is definitely not the "most appropriate".

Answer: According to the reviewer’s remark this sentence was rewritten

  1. In line 149 the authors use the expression: "... shown GOOD EeAChE inhibition...". Question: How can the authors consider as "good" a behaviour (measured by IC50) where there is a 1000-fold difference? IC50donepezil = 20.8 nM vs IC50 4f = 12.6 µM? Firstly, there is a lack of statistical evidence to be sure of that and secondly, the word "good" is too subjective to interpret scientific results (it seems to be more an opinion of the authors on their own results). I strongly recommend rewriting those lines. The same goes for lines 136, 154 and 195, where that word is used.

Answer: According to the reviewer's comments, the sentences in which the word "good" is used have been rewritten.

  1. In line 164 the authors mention "these molecules seem very promising, compared to the sulfonamides recently described in the literature...". This type of expression (which is subjective) should go at the end of the article and not in the first results section, as not all the findings are mentioned and their importance is not yet discussed. I recommend removing it or moving it to the "Discussion" section.

 Answer: According to the reviewer's comments, the paragraph was moved to the conclusion section.

  1. At the end of section 2.2.3 (antioxidant capacity assay, line 203), the authors arbitrarily choose to evaluate the Nrf2 transcriptional activation potencies of only 3 molecules (out of the total of 8 with which the study started). Why were the others not tested? Please give an appropriate argument for your exclusion decision.

 Answer: Given the high cost of the NrF2 test and to make judicious use of our resources, we did not test all compounds. However, the selection of compounds we tested was made in a rational manner. Specifically, we focused on compounds 4a, 4d, and 4f, which demonstrated a multi-target profile as the only ones exhibiting activity as both calcium channel blockers and cholinesterases.

A clarification on this aspect has been added to the manuscript

  1. In line 227 the word "powers" should be replaced by "potencies" or "capacities".

Answer: Ok, Corrected

Reviewer 2 Report

The manuscript reported the MTDLs 4a and 4f showing simultaneously cholinesterase 22 inhibition, calcium channel blockade, antioxidant capacity and Nrf2-ARE activating effect. The manuscript can be published in the journal with minor revision.

1. The compounds are solid or liquid?

2.Line 435: whereas compound 4d showed dual inhibition of both ChEs. Here compound 4d should be compound 4f?

3.The interaction between the compounds 4a and 4f can be studied by docking those compounds with ChE.

4. The IC50 of donepezil for BChE and the IC50 for tacrine should be given in table 1.

Minor editing of English language required.

Author Response

Reviewer 2

The manuscript reported the MTDLs 4a and 4f showing simultaneously cholinesterase 22 inhibition, calcium channel blockade, antioxidant capacity and Nrf2-ARE activating effect. The manuscript can be published in the journal with minor revision. 

  1. The compounds are solid or liquid?

Answer: Thanks to the reviewer for this comment, the compounds are solid and this information has been added in the manuscript:

2.Line 435: whereas compound 4d showed dual inhibition of both ChEs. Here compound 4d should be compound 4f?

Answer: This error has been corrected

3.The interaction between the compounds 4a and 4f can be studied by docking those compounds with ChE.

Answer: Thanks to the reviewer for this interesting comment, consequently a docking study was performed for compound 4a and 4f against cholinesterase.

  1. The IC50of donepezil for BChE and the IC50for tacrine should be given in table 1.

Answer: The IC50 of donepezil for BChE and the IC50 for tacrine were added

Round 2

Reviewer 1 Report

Dear Authors,

I appreciate your thoughtful attention in addressing the points raised during my initial review and revising your manuscript accordingly.

Upon thorough re-evaluation, I am pleased to see that all the issues previously mentioned have been adequately resolved. Given these revisions, I am of the opinion that the manuscript is now in a state suitable for publication.
